# A miR-9-5p/FOXO1/CPEB3 Feed-Forward Loop Drives the Progression of Hepatocellular Carcinoma

**DOI:** 10.3390/cells11132116

**Published:** 2022-07-05

**Authors:** Hui Hu, Wei Huang, Hong Zhang, Jianye Li, Qiong Zhang, Ya-Ru Miao, Fei-Fei Hu, Lu Gan, Zhenhong Su, Xiangliang Yang, An-Yuan Guo

**Affiliations:** 1Center for Artificial Intelligence Biology, Hubei Bioinformatics & Molecular Imaging Key Laboratory, Key Laboratory of Molecular Biophysics of the Ministry of Education, College of Life Science and Technology, Huazhong University of Science and Technology, Wuhan 430074, China; huhui@hust.edu.cn (H.H.); transomics@gmail.com (Q.Z.); miaoyr@hust.edu.cn (Y.-R.M.); hufeifei@wust.edu.cn (F.-F.H.); 2National Engineering Research Center for Nanomedicine, College of Life Science and Technology, Huazhong University of Science and Technology, Wuhan 430074, China; huangwei2020@whut.edu.cn (W.H.); jianye_li@foxmail.com (J.L.); lugan@hust.edu.cn (L.G.); 3School of Chemistry, Chemical Engineering and Life Sciences, Wuhan University of Technology, Wuhan 430070, China; 4Department of Gastroenterology, Wuhan Third Hospital, Wuhan 430060, China; yumojunya1518@163.com; 5Hubei Key Laboratory for Kidney Disease Pathogenesis and Intervention, Medical College, Hubei Polytechnic University, Huangshi 435000, China; szh7977@163.com

**Keywords:** hepatocellular carcinoma, regulatory networks, feed-forward loop, prognosis, mechanisms

## Abstract

Hepatocellular carcinoma (HCC) is the third leading cause of cancer-related death worldwide, but its regulatory mechanism remains unclear and potential clinical biomarkers are still lacking. Co-regulation of TFs and miRNAs in HCC and FFL module studies may help to identify more precise and critical driver modules in HCC development. Here, we performed a comprehensive gene expression and regulation analysis for HCC in vitro and in vivo. Transcription factor and miRNA co-regulatory networks for differentially expressed genes between tumors and adjacent tissues revealed the critical feed-forward loop (FFL) regulatory module miR-9-5p/FOXO1/CPEB3 in HCC. Gain- and loss-of-function studies demonstrated that miR-9-5p promotes HCC tumor proliferation, while *FOXO1* and *CPEB3* inhibit hepatocarcinoma growth. Furthermore, by luciferase reporter assay and ChIP-Seq data, CPEB3 was for the first time identified as a direct downstream target of FOXO1, negatively regulated by miR-9-5p. The miR-9-5p/FOXO1/CPEB3 FFL was associated with poor prognosis, and promoted cell growth and tumor progression of HCC in vitro and in vivo. Our study identified for the first time the existence of miR-9-5p/FOXO1/CPEB3 FFL and revealed its regulatory role in HCC progression, which may represent a new potential target for cancer therapy.

## 1. Introduction

MicroRNAs (miRNAs) and transcription factors (TFs) play crucial roles in multiple biological processes and participate in the development of human diseases including cancers [1,2]. They can regulate each other and co-regulate the same target gene to form feed-forward loops (FFLs) involved in many biological processes and diseases [3]. Previous studies have revealed the role of miR-19/CYLD/NFKB and miR-429/MYCN/MFHAS1 FFLs in the development or relapse of T-lineage acute lymphoblastic leukemia (T-ALL) [4,5]. FOXP3-miR-7/miR-155-SATB1 FFL has been reported to prevent the transformation of healthy mammary epithelium into a cancerous phenotype [6]. Src/Sox2/miR-302b FFL increased metastatic progression of breast cancer [7]. In addition, our previous projects revealed key factors or FFL module involvement in promoting cancer cell stemness [8] and CD19-specific chimeric antigen receptor (CAR) T-cell immunotherapy [9]. To detect key factors or FFL modules in biological processes and diseases, we developed an online analysis server FFLtool [10].

The high prevalence and mortality rate of hepatocellular carcinoma (HCC) make it the main health burden at present [11]. These challenges give rise to the urgent need to systematically investigate the pathogenesis and identify potential biomarkers for HCC. In the last decade, various studies have shown that the abnormal expression of miRNAs or TFs is broadly associated with the pathogenic mechanism of HCC. For example, miR-206 is a powerful tumor suppressor that regulates cell-cycle progression of HCC [12]. The miR-449 family inhibits cell migration by targeting SOX4 [13]. In addition, circular RNA circMTO1 suppresses HCC progression by down-regulating the target of oncogenic miR-9 [14]. FOXO1 is a TF that acts as a tumor suppressor in regulating cell cycle, progression, differentiation, metabolism and survival [15,16]. Recent evidence has suggested that miR-96 promotes cell growth and migration by inhibiting FOXO1 in HCC [17]. Up-regulated SOX12 promotes HCC invasion and metastasis [18]. Furthermore, HNF4A, TP53, STATs and NFKB have also been reported as key TFs related to the development of HCC [19]. Although many TFs and miRNAs have been reported to be important in HCC, their co-regulation and FFL modules in HCC have not been studied. An improved understanding of these matters may help to identify novel driver modules in HCC development.

In this study, to identify novel driver modules of HCC, we performed network analysis based on FFLs formed by miRNA–TF. Combined with bioinformatic analysis and in vitro and in vivo experiments, we validated the presence of key FFLs and their important regulatory roles in the proliferation of HCC. Our findings provide clues for HCC pathogenesis and candidate biomarkers.

## 2. Materials and Methods

### 2.1. Data Source

RNA sequencing (RNA-seq) (V2) and miRNA sequencing (miRNA-seq) data (level 3) of 374 HCC cases were downloaded from The Cancer Genome Atlas (TCGA) data portal (https://portal.gdc.cancer.gov/, accessed on 6 May 2017). Expression data (RSEM normalized (TCGA normalized mRNA expression)) of paired cancer and adjacent tissue samples (50 paired for RNA-Seq, 49 paired for miRNA-Seq) were used to analyze differentially expressed genes (DEGs) and differentially expressed miRNAs (DEMs).

### 2.2. Differentially Expressed Genes

The DEGs in our study were carried out by the Bioconductor package NOISeq [20]. Gene expression higher than 20 normalized RSEM count, and miRNAs above 10 normalized RSEM count in any sample were retained for differential analysis. The cutoffs for DEGs were prob >0.99 and |Fold Change (FC)| ≥2. TFs and miRNAs are regulatory factors with amplificatory effects for their functions; we set prob >0.99 and |Fold Change| ≥1.6 as the cutoffs of differentially expressed TFs or differentially expressed miRNA (DEMs) (Appendix A). All heatmaps were normalized by genes in a row and hierarchically clustered through Euclidean distance using the R pheatmap package.

### 2.3. Enrichment Analysis

Gene ontology (GO) and Kyoto Encyclopedia of Genes and Genomes (KEGG) pathway enrichment analysis of all DEGs were conducted by tool Metascape (http://metascape.org/gp/index.html, accessed on 11 November 2018) (*p* ≤ 0.05) [21]. Functional enrichment analysis of DEMs was performed by the DIANA-miRPath version (v.)3.0 [22]. All results were plotted by R package ggplot2 (https://ggplot2-book.org/, accessed on 15 December 2018).

### 2.4. Survival Analysis

Kaplan–Meier survival analysis was performed based on 332 tumor samples of TCGA HCC data. There were 368 samples for TCGA HCC data of which 34 had no clinical follow-up information, and were removed in survival analysis. For each gene, low- and high-expression samples were defined by expression levels of the given gene lower than 25th percentile and higher than 75th percentile, respectively, giving 83 samples per group. For multi-gene survival analysis, samples were divided into low- and high-expression groups by median gene expression levels. Given that low expression of miR-9-5p, high expression of *FOXO1*, and high expression of *CPEB3* have good prognosis, respectively, we compared the best prognostic subgroups (low miR-9-5p and high *FOXO1* and high *CPEB3*) with the worst prognostic subgroups (high miR-9-5p and low *FOXO1* and low *CPEB3*). Statistical significance (*p*-value) was calculated by log-rank (Mantel–Cox) test. Data was removed when the patients were lost to follow-up.

### 2.5. Regulatory Networks and Hub Analysis

We constructed miRNA–TF co-regulatory networks for all DEGs and DEMs, based on interaction data collected in previous work [3,23], including experimental and predicted TF–target/miRNA and miRNA–target/TF interaction pairs. All networks were visualized by Cytoscape (version 3.4.0). To determine hub nodes (TFs, miRNAs and genes) in networks, we calculated eight centrality feature values of network nodes, based on CytoNCA [24]. A higher value meant the node was more important in the network. We retained the top 50% of genes for each centrality feature value, and then obtained 30 hub nodes (including 10 miRNAs, 7 TFs and 13 genes) in the network by the intersection of eight features. We then retained key nodes with high expression and large differences between cancer and adjacent samples, representing key edges in the network with reliable evidence (Chromatin Immunoprecipitation sequencing (ChIP-Seq) experiments or TargetScan prediction) to support their regulatory relationships. Next, genes and TFs whose expression was not affected by DNA methylation and copy number variation in HCC were selected as final candidates for key FFLs.

### 2.6. Cell Lines and Cell Culture

Human hepatocellular carcinoma cell line HepG2 was purchased from Cell Bank, Chinese Academy of Sciences. Human hepatocarcinoma cell line Bel7402 was purchased from China Center for Type Culture Collection (CCTCC, Wuhan, China). Human liver cell line HL7702, and human hepatocarcinoma cell lines Huh7 and LM3 were kindly provided by Prof. Bixiang Zhang [25]. Cells were cultured on plastic plates in DMEM (high glucose Dulbecco’s modified Eagle’s medium) with 10% fetal bovine serum (Life Technologies, Carlsbad, CA, USA), and 100 mg/mL penicillin/streptomycin (Gibco) at 37 °C with 5% CO_2_. The cells were randomly assigned to each experimental group and the potential presence of mycoplasma was monitored via continuous microscopic imaging.

### 2.7. Transfection

Cells were transfected with small interfering RNA (siRNA) using Lipofectamine 2000 (Life Technologies, Carlsbad, CA, USA) according to manufacturer’s instructions. The negative control siRNA, siRNA duplexes and shRNA were synthesized from Genephama Biotech (Suzhou, China). All the siRNA and shRNA sequences are listed in Appendix A. The miR-9-5p mimic, specific inhibitor molecules, and appropriate negative control molecules were purchased from Genephama Biotech (Suzhou, China). All the mimic, inhibitor, and negative control molecules were transfected using X-treme GENE siRNA Transfection Reagent (Roche, Mannheim, Germany), based on the manufacturer’s protocol. The *FOXO1* overexpressed plasmids were supplied by Prof. Lu Gan, having been generated by subcloning PCR-amplified full-length human *FOXO1* cDNA (including endogenous 3′UTR). The *CPEB3* expression construct was generated by subcloning PCR-amplified full-length human *CPEB3* cDNA into the pBABE (Plasmid #21836, addgene).

### 2.8. Invasion Assay

The upper chambers were coated with 50 μg extracellular matrix gel (Corning, Corning, NY, USA). After incubation for 24–48 h, the invaded cells on the lower membrane were stained with 4% paraformaldehyde, then stained with 0.1% crystal violet. The upper cells were moved gently by the soft medical cotton ball. For each group, 10 pictures were randomly taken under a microscope with 400× magnification (Olympus, Tokyo, Japan). Cell numbers per field were calculated by ImageJ.

### 2.9. Cell Proliferation Assay

Cell proliferation was measured with the Cell Counting Kit-8 (CCK-8) assay (Dojindo Corp., Kumamoto, Japan). HepG2 cells or Bel7402 cells were seeded in 96-well plates (8 × 10^3^ cells/well). Vectors or RNAi were transfected for 24 h (HepG2) and 48 h (Bel7402). 10 µL CCK-8 reagent was added to each well. After incubating for 4 h, the cell density was measured indirectly through quantification of the solubilized formazan product at 450 nm using Multiskan GO (ThermoFisher, Waltham, MA, USA). Three independent experiments were performed.

### 2.10. Cell Apoptosis Assay

The Annexin V-FITC/PI apoptosis detection kit was purchased from Zoman Biotechnology Co. Ltd. (Beijing, China). Bel7402 cells were transfected with vector, miR-9-5p inhibitor, FOXO1, and CPEB3 for 48 h. Then, cells were harvested and washed with cold PBS. After 1500 rpm 5 min centrifuge, cells were resuspended in 500 μL binding buffer. Then, 5 μL of Annexin V-FITC and 10 μL of PI were added and incubated with the cells for 15 min in the dark. Finally, the stained cells were analyzed by CytomicsTM FC 500 (Beckman Coulter, Brea, CA, USA). The apoptosis ratio was counted by the percentage of AnnexinV-FITC-positive cells, both PI-negative and PI-positive.

### 2.11. Animals

Four-week old BALB/c nude male mice were obtained from Beijing Vital River Laboratory Animal Technology Co., Ltd. (Beijing, China). The mice were randomly assigned to the control or treated group, each group had six mice. The experimental technicians were blinded to the expected outcome of the treatment. All animals received humane care in compliance with the Principles of Laboratory Animal Care formulated by the National Society of Medical Research and the US National Institutes of Health guidelines. The protocol was approved by the Institutional Animal Care and Use Committee of Huazhong University of Science and Technology (Number: 2808 and 1 June 2018 of approval).

Xenograft tumor-bearing mice were constructed by subcutaneous injection of HepG2 cells (0.5~1 × 107 cells per mouse) into the flanks of male BALB/c nude mice. The Bel7402 cells or HepG2 cells were planted on a 10 cm cell-cultured plate (Corning, Corning, NY, USA) until the cells were 80% full, then transfected with inhibitor NC, miR-9-5p inhibitor through the Roche X-treme GENE SiRNA Transfection Reagent. The empty vector, CPEB3 overexpression plasmid, CPEB3 shRNA, and FOXO1 overexpression plasmid were transfected with Lipofectamine 2000 (Life Technologies, Carlsbad, CA, USA). Cells were harvested after being transfected for 24 h. Then, after a week, we measured the length and width of the tumor every two days, and tumor volume was calculated by V = 1/2 × L × W2.

### 2.12. Statistical Analysis

All data were described as the mean ± s.e.m and performed in triplicate at least. Comparisons between two groups for tumor weight were performed using Student’s two-sided *t*-test. For multiple group comparison, significant differences between the means of control versus treatment groups (including quantitative real-time PCR (qRT-PCR), cell viability, cell number and apoptosis) were tested by one-way analysis of variance (ANOVA), followed by Dunnett’s post hoc test for multiple comparisons. One-way ANOVA testing was used for intergroup comparison of tumor volume. Statistical analysis was accomplished by GraphPad Prism 8.0 and R 4.1.0. Single, double, triple and quadruple asterisks indicate statistical significance: * *p* < 0.05, ** *p* < 0.01, *** *p* < 0.001; **** *p* < 0.0001; while ns indicates a non-significant result. The detailed procedures of analyses, including high-throughput sequencing data, luciferase reporter gene assay, chromatin immunoprecipitation, real-time qPCR analysis, and western blotting assays are presented in Appendix A.

## 3. Results

### 3.1. The Gene Function Analysis and Regulatory Network Analyses Revealed miR-9-5p/FOXO1/CPEB3 FFL in HCC

To study gene expression and regulation in HCC, RNA-Seq and miRNA-Seq data (level 3) of 50 paired HCC tumor and adjacent tissue samples from TCGA were analyzed. As a result, 387 significantly differentially expressed genes (DEGs) and 17 differentially expressed miRNAs (DEMs) were detected in the comparison of tumors with adjacent samples (Appendix A). About 8% (32) of DEGs were TFs, as shown in Figure 1A. Functional enrichment analysis revealed that the largest quantity of upregulated DEGs were involved in cell cycle, oocyte meiosis and pathways in cancer, while most down-regulated DEGs were associated with the cytokine-cytokine receptor, drug metabolism, and HTLV-I infection pathways, which may suggest their roles in the development of HCC (Figure 1B). For example, LYVE1, a down-regulated gene in HCC, may constitute an early biomarker of postoperative survival in HCC patients [26]. Meanwhile, VCAN, which was upregulated in HCC, could serve as a potential biomarker for early-HCC diagnosis [27]. Besides, target genes of DEMs were mainly enriched in the processes of cancer, including glioma, hippo signaling pathway, and signaling pathways regulating pluripotency of stem cells (Appendix A).

Next, to explore the mechanism of gene expression regulation in HCC, we built miRNA–TF gene regulatory networks based on the DEGs and DEMs mentioned above. There were 118 nodes (17 TFs, 13 miRNAs, and 88 genes) with 481 edges in the network (Figure 1C). Based on 30 hub nodes (7 TFs, 10 miRNAs, and 13 genes) detected by CytoNCA [24], a sub-regulatory network (30 hub nodes with 138 edges) was constructed (Appendix A). Hub nodes, which regulate many genes or are regulated by numerous factors, may play important roles in the development of liver cancer. For example, hub node gene CPEB3 has been described as a newly discovered tumor suppressor in HCC [28], and hub TF FOXO1 was responsible for blocking HCC proliferation [29]. In our results, the hub regulator miR-9-5p was the most significant DEM (log2FoldChange = 4.012) between tumor and adjacent samples (Appendix A), which was reported to promote cell proliferation of HCC [30]. The sub-network of miR-9-5p consisted of 16 genes and 14 TFs (Figure 1D). Interestingly, as target genes of miR-9-5p in the network, both *FOXO1* and *CPEB3* were down-regulated and reported to be associated with HCC (Figure 1E), and their expression levels were not affected by DNA methylation or copy number variation (Appendix A). In addition, *CPEB3* was regulated by FOXO1 in our regulatory network, thus forming a feed-forward loop (FFL) with FOXO1 and miR-9-5p (Figure 1E). Although miR-9-5p, CPEB3, and FOXO1 in HCC have been studied alone and together, the FFL consisting of these three elements has not been studied and may be more powerful in explaining HCC disease progression. Thus, we focused on the role in HCC progression of the FFL formed by FOXO1, miR-9-5p and CPEB3.

### 3.2. miR-9-5p Promotes Tumor Proliferation, while FOXO1 and CPEB3 Play Tumor-Suppressive Roles in HCC

As shown in Figure 1E, miR-9-5p had higher expression in tumor tissues compared to adjacent tissues, while *FOXO1* and *CPEB3* were expressed higher in adjacent tissues. To further investigate the roles of miR-9-5p, FOXO1 and CPEB3 in HCC, we detected their expression in several HCC cell lines and liver cell line HL7702 (Figure 2A). We found that miR-9-5p expression was elevated in HepG2, Bel7402 and Huh7, compared to liver cell line HL7702. The expression levels of *FOXO1* and *CPEB3* were significantly decreased in HepG2, Huh7, Bel7402, and LM-3 cells. To further explore their biological roles in HCC, we performed gain- and loss- of-function experiments for Bel7402 and HepG2 cell lines (Figure 2B,C and Appendix A). Significantly increased expression of miR-9-5p or *FOXO1* or *CPEB3* was observed in Bel7402 cells treated with miR-9-5p mimics or *FOXO1* vector or *CPEB3* vector, respectively, compared with the control group of qRT-PCR, whereas silencing these three factors resulted in their decreased expression (Appendix A). The inhibition of miR-9-5p or overexpression of *FOXO1* and *CPEB3* significantly suppressed cell growth (Figure 2B and Appendix A), decreased cell invasion ability in vitro as seen by transwell assay (Figure 2C), and significantly enhanced cell apoptosis of Bel7402 and HepG2 cells (Figure 2D and Appendix A). Conversely, overexpressed miR-9-5p or inhibition of *FOXO1* and *CPEB3* markedly promoted cell proliferation (Figure 2B and Appendix A), which was also confirmed by the in vitro transwell assay results (Figure 2C). Furthermore, co-transfection of *FOXO1* or *CPEB3* vector and siRNA significantly reduced their inhibitory effects on cell proliferation in HCC (Figure 2B and Appendix A).

In summary, miR-9-5p, FOXO1 and CPEB3 may affect progression of HCC through impact cell proliferation and invasion. Past efforts have demonstrated that miR-9-5p and FOXO1 were involved in the progression of HCC in vivo [14,15,30] and CPEB3 silencing increased tumor size and weight in HCC [28]. In this work, we observed that the tumor-formation ability of the CPEB3 overexpressing cells was reduced in nude mice when compared with control (Figure 2E). These results suggest that miR-9-5p promotes HCC cell proliferation and invasion in vitro, whereas FOXO1 and CPEB3 have the opposite effect.

### 3.3. The Existence of FFL among miR-9-5p, FOXO1 and CPEB3 in HCC

Furthermore, we aimed to study the relationships between miR-9-5p, FOXO1 and CPEB3, and prove the existence of the FFL that they form. Firstly, qRT-PCR results in Bel7402 cells indicated the FOXO1 acts as a negative regulator of miR-9-5p and possibly a positive modulator of CPEB3 (Figure 3A). *FOXO1* expression was significantly positively associated with *CPEB3* (Appendix A; Correlation coefficient = 0.34, *p* = 7.02 × 10^−11^), and *CPEB3* expression was significantly reduced by *FOXO1* silencing, although *CPEB3* expression was not altered after *FOXO1* overexpression. *CPEB3* expression was significantly suppressed with miR-9-5p upregulation in Bel7402 cells, whereas *CPEB3* expression in the miR-9-5p inhibitor group showed a contrary result (Figure 3A). Because FOXO1 was reported to be a target gene of miR-9-5p by luciferase reporter and western blot [31], we next verified whether CPEB3 is a direct target of miR-9-5p. Two loci on the 3′ UTR of *CPEB3* were predicted as potential binding sites of miR-9-5p through TargetScan algorithm, named CPEB3−1 and CPEB3−2, respectively (Figure 3B). The luciferase activities of wild-type CPEB3−1 (CPEB3−1 WT), wild-type CPEB3−2 (CPEB3−2 WT) and mutant-type CPEB3−2 (CPEB3−2 Mut) were significantly inhibited (*p* < 0.05) by transfecting miR-9-5p, while that of mutant-type CPEB3−1 (CPEB3−1 Mut) hardly changed (Figure 3C). These may indicate that miR-9-5p downregulated *CPEB3* expression by directly targeting the locus 1 (position 437-443) of 3′ UTR of *CPEB3*, rather than locus2 (position 1367-1373).

Next, ChIP-Seq analysis was employed to evaluate whether FOXO1 directly regulates *CPEB3* transcription. ChIP-Seq with FOXO1 antibodies showed strong peaks of FOXO1 binding associated with the two *CPEB3* promoters in HepG2 cells (Figure 4A). Furthermore, the result by ChIP-qPCR demonstrated that FOXO1 binding was significantly enhanced after immunoprecipitation at *CPEB3* transcript isoform 2 (NCBI: NM_001178137.1) compared with IgG, but *CPEB3* transcript isoform 1 (NM_014912.5) was only slightly elevated (Figure 4B). In addition, CPEB3 expression was remarkably upregulated after FOXO1 overexpression, both at gene (Figure 4C) and protein expression levels in HepG2 cells (Figure 4D). Also, CPEB3 promoter reporter assay showed that FOXO1 significantly enhanced CPEB3 promoter activity (Figure 4E) through each of the three binding sites in Figure 4A. These findings all indicated that FOXO1 may be a positive modulator of CPEB3. Furthermore, FOXO1 overexpression affected not only CPEB3 expression but also a huge number of pathways. Functional enrichment analysis demonstrated that DEGs of *FOXO1* overexpression compared with control (Appendix A) were associated with non-alcoholic fatty liver disease, FoxO signaling pathway, cell cycle, and autophagy pathways, closely related to the occurrence and development of HCC (Figure 4F). It was found that FOXO1 or CPEB3 overexpression could reverse the promotion of cell migration and invasion by miR-9-5p [31,32]. These results suggest regulatory relationships between miR-9-5p, FOXO1 and CPEB3, and further reveal the existence of miR-9-5p/FOXO1/CPEB3 FFL (Figure 1E).

### 3.4. The miR-9-5p/FOXO1/CPEB3 FFL May Promote Progression of HCC In Vivo

To investigate the miR-9-5p/FOXO1/CPEB3 FFL effects in HCC patients, we performed Kaplan–Meier survival analysis for one, two and three genes combinations in miR-9-5p/FOXO1/CPEB3 FFL. Clinical characteristics distribution of all patients can be seen in Appendix A. Results indicated that the downregulations of *FOXO1* and *CPEB3* were significantly correlated with better prognosis, while high expression of miR-9-5p was associated with poor prognosis (Figure 5A and Appendix A). Patients with the combination of low miR-9-5p and high *FOXO1* had significantly improved overall survival at 60 months (*p* = 0.0092, Figure 5A) and 100 months (*p* = 0.0152, Appendix A) versus the opposite group. Similar to the combinations of low miR-9-5p and high *CPEB3*, or high *FOXO1* and high *CPEB3* group, the survival was longer than their respective corresponding opposite groups (60 months: *p* = 0.0002; *p* = 0.0153, Figure 5A, 100 months: *p* = 0.0024; *p* = 0.0392, Appendix A). Likewise, patients with low miR-9-5p, high *FOXO1* and high *CPEB3* had significantly longer overall survival than the opposite expression group (*p* = 0.009 for 60 months, Figure 5A, *p* = 0.018 for 100 months, Appendix A).No difference in overall survival was observed between patients with combinations of high miR-9-5p and high *FOXO1*, or high miR-9-5p and high *CPEB3*, or high *FOXO1* and low *CPEB3*, versus the opposite groups at 60 months (Appendix A) and 100 months (Appendix A). However, high miR-9-5p, high *FOXO1* and low *CPEB3* had significantly shorter overall survival than the opposite expression group (*p* = 0.018 for 60 months, Appendix A, *p* = 0.025 for 100 months, Appendix A).

To further explore the function of miR-9-5p/FOXO1/CPEB3 FFL in vivo, we built hepatoma xenografts in nude mice. Bel7402 cells transfected or co-transfected with empty vector, miR-9-5p inhibitor, *FOXO1* overexpression plasmid, and *CPEB3* shRNA were injected subcutaneously into the flank of each nude mouse (Appendix A). As expected, tumor volumes (after day 17) and weights were significantly reduced with miR-9-5p silencing or FOXO1 overexpression, and tumor suppression was sustained after miR-9-5p silencing along with FOXO1 overexpression (Figure 5B,C,E). Next, qRT-PCR analysis indicated that miR-9-5p expression was significantly decreased in tumor tissues after treatment with the combination of miR-9-5p inhibitor/*FOXO1*/*CPEB3*-shRNA (Figure 5G, left). *CPEB3* expression increased markedly with miR-9-5p silencing and slightly increased with *FOXO1* overexpression (Figure 5G, right). However, tumors grew faster when CPEB3 was silenced (Figure 5D) and the tumor-promoting action was not suppressed by miR-9-5p silencing and FOXO1 overexpression (Figure 5F). This may be because alterations in downstream effectors have a greater impact on tumor progression compared to upstream regulators. These findings all indicate that the miR-9-5p/FOXO1/CPEB3 FFL may promote progression of HCC in vivo through downregulation of effector gene CPEB3 (Figure 5).

## 4. Discussion

HCC is an important worldwide cause of cancer-related mortality [33]. The poor prognosis of liver cancer and its high recurrence ratio [34] make it imperative to investigate the mechanisms and biomarkers of HCC [35]. With the fact that many TFs and miRNAs play important roles in liver cancer and the successful application of FFL analysis in disease mechanisms [4,35], a novel avenue has been opened to investigate the molecular mechanisms of HCC pathogenesis. The TF-miRNA co-regulation network constructed, based on DEGs and DEMs, aids the identification of potential functional regulators in HCC. In this research, we detected a key FFL module through bioinformatics analyses; the miR-9-5p/FOXO1/CPEB3 FFL was further verified to contribute to the progression of HCC in vivo and in vitro.

Our study first revealed the existence of miR-9-5p/FOXO1/CPEB3 FFL in HCC (Figure 1E). We demonstrated that miR-9-5p was highly expressed in HCC, and formed an FFL with down-regulated FOXO1 and CPEB3 (Figure 1E,F and Figure 2A). The expression of these three factors has been extensively validated by qRT-PCR or WB and IHC in human HCC tissues [36,37,38]. Further, given the expression of *FOXO1* or *CPEB3* was not affected by DNA methylation or mutations in TCGA HCC (Appendix A), we speculated that expression changes of the two genes were mainly due to their regulation. CPEB3 was confirmed to be a functional target gene of miR-9-5p by luciferase reporter (Figure 3C), which is consistent with miR-9-5p targeting binding to CPEB3 [32]. We identified that FOXO1 directly regulates *CPEB3* transcription by ChIP-Seq assays, Western blot (Figure 4A,D) and luciferase reporter in HepG2 cells (Figure 4E), although the regulation was not significant in Bel7402 cells (Figure 3A) and tumor tissues (Figure 5G, right). In addition, luciferase reporter confirmed that FOXO1 is a target gene of miR-9-5p [31]. Interestingly, experimental studies showed that *FOXO1* may in return have downregulated miR-9-5p (Figure 3A and Figure 5G). In line with that, 3′ UTR of FOXO1 may function as a competing endogenous RNA (ceRNA) to regulate miR-9-5p activity [39]. Therefore, we confirmed regulatory relationships among the three factors and the existence of miR-9-5p/FOXO1/CPEB3 FFL.

Integrative analysis of sequencing data and clinical data including patient survival and pathological grade can improve clinical application. Results showed that low expression of *FOXO1* and *CPEB3* or high expression of miR-9-5p correlated with poor prognosis in HCC (Figure 5A and Appendix A), similar to two- or three-factor combination groups of low *FOXO1*, low *CPEB3* and high miR-9-5p (Figure 5A and Appendix A). Given the high expression of miR-9-5p, low expression of *FOXO1*, and low expression of *CPEB3* in HCC data (Figure 5A), this implied that the miR-9-5p/FOXO1/CPEB3 FFL could be a poor prognostic biomarker for HCC. The expression analysis of pathological stages showed that only *FOXO1* expression was significantly associated with tumor stage, while miR-9-5p and *CPEB3* gene expression were less associated with clinical stage (Appendix A), implying that FOXO1 may have a potential indicative role in pathological grading. Gain- and loss-of-function analyses indicated that high expression of miR-9-5p was associated with aggressive tumor phenotypes and poor clinical diagnosis (Figure 2B,D and Figure 5A), while FOXO1 and CPEB3 inhibited tumor proliferation (Figure 2B–D). Furthermore, FOXO1 or CPEB3 overexpression was also found to reverse the promotive effect of miR-9-5p on cell migration and invasion [31,32], implying an important role for this FFL in the development of HCC. Similar to in vitro results, animal experiments indicated that miR-9-5p silencing and FOXO1 overexpression could suppress the growth of hepatoma xenografts (Figure 5B,C,E), while silenced CPEB3 promoted tumor growth in vivo (Figure 5B,D), which was consistent with previous reports for HCC [15,28,30]. However, co-transfection of miR-9-5p inhibitors, *FOXO1* overexpression plasmid, and *CPEB3* shRNA significantly increased the growth of hepatoma xenografts, similar to the effect of transfection with *CPEB3* shRNA alone (Figure 5B), suggesting that in the case of CPEB3 silencing, the effects of promoting CPEB3 expression by miR-9-5p silencing and FOXO1 overexpression were masked. This implies that the expression change of downstream effector gene in the FFL maybe has a greater effect on cancer than the upstream regulators. In conclusion, we speculated that miR-9-5p/FOXO1/CPEB3 FFL may promote the progression of HCC due to the downregulation of the effector gene CPEB3.

Given the importance of miR-9-5p/FOXO1/CPEB3 FFL in HCC, we tried to explore the downstream pathways of this regulatory loop. CPEB3, cytoplasmic polyadenylation element binding protein 3, acted as the downstream effector gene of the FFL involved in downstream pathways. CPEB3 promoted human HCC cell proliferation and metastasis [28] and was downregulated in HCC [40]. Given that EGFR has been reported as a downstream gene of CPEB3 in HCC [28], we investigated whether other genes that interact with EGFR (BioGRID database) were also downstream genes of CPEB3. Through Western blotting, we demonstrated that CPEB3 induced upregulation of NCK2 and LIMS1 (Appendix A). NCK2 is an Nck-related adaptor protein involved in growth factor receptor kinase signaling pathways, which was found to interact with LIMS1 to participate in ILK signaling and downregulate EGFR protein [41,42]. EGFR is correlated with poor prognosis, and cancer metastasis in different cancers including HCC [28,43], and it was negatively regulated by CPEB3 in HCC [28]. It has previously been shown that FOXO1 activation suppressed the ILK pathway [44], and the carcinogenic ability of ILK in HCC cells has been confirmed in vivo [45]. Furthermore, RSU1, a suppressor of Ras-dependent oncogenic transformation, was reported to interact with LIMS1, and was attributed to inhibiting cell proliferation and invasion in tumor cells including HCC [46,47]. Based on information derived from the literature and original data from this work, miR-9-5p/FOXO1/CPEB3 FFL may facilitate HCC pathogenesis through the EGFR and ILK signaling pathways (Figure 6). However, future in-depth studies of these downstream mechanisms are needed.

Although our study has proposed a novel functional miR-9-5p/FOXO1/CPEB3 FFL in the tumor growth of HCC, more complete studies are needed in future experiments to enhance the validation of the functional role of the FFL. The potential effects of intervening regulators in the FFL for therapy and preventing HCC progression will be a further research direction. Effects on invasion and metastasis after co-transfection with multiple genes of the FFL may be studied, and the reversion experiments for the expression of regulatory loop genes need to be subsequently validated in mouse tumor models. We suggest that the downstream signaling pathways may be affected by the regulatory loop, but further investigation is needed to elucidate the detailed mechanisms of how the FFL affects HCC development. In addition, the complex regulatory network system in HCC implies a possible inter-regulation and influence between different FFLs (Figure 1C), which is difficult to eliminate and resolve, and is a shortcoming of the study. This study revealed key factors and the vital FFL, and explored their association with survival data of HCC. However, more clinical data and experimental validation of key factors are needed to enhance the credibility of their practical application in clinical studies. Despite some shortcomings, this project provides evidence and a theoretical basis for the pathogenic mechanism and potential molecular markers of HCC.

In summary, we performed miRNA–TF gene regulatory network analysis to investigate the possible mechanisms for HCC. Functions in carcinogenesis of three hub nodes including miR-9-5p, FOXO1 and CPEB3 were investigated. CPEB3 was identified as a direct target of FOXO1. Most importantly, we demonstrated the existence of miR-9-5p/FOXO1/CPEB3 FFL and validated its regulation role in HCC tumor proliferation in vivo and in vitro. This study provides us with a deeper understanding of the transcriptional and post-transcriptional regulatory mechanisms underlying HCC progression, which will provide a new perspective for further understanding the pathogenesis of HCC, and the selection of diagnostic and therapeutic targets.

## Figures and Tables

**Figure 1 cells-11-02116-f001:**
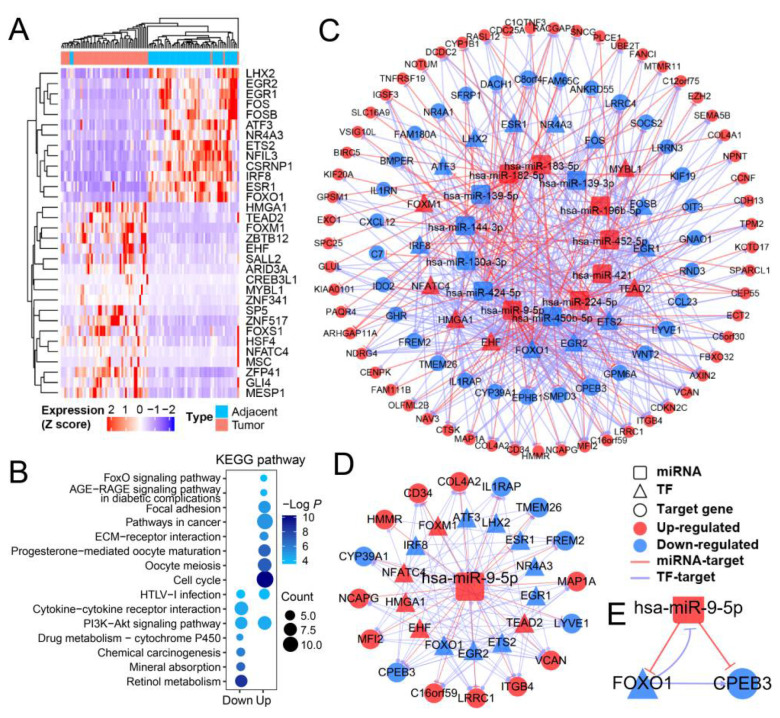
The miRNA–TF gene regulatory network and hub nodes in HCC. (**A**) Heatmap for differentially expressed TFs in HCC tumor vs. adjacent tissues. The color gradient from blue to red indicates the scaled expression (Z score) from low to high. (**B**) Result of KEGG pathway enrichment analysis for differentially expressed genes (DEGs). (**C**) miRNA–TF–gene FFL regulatory network for all DEGs and differentially expressed miRNAs (DEMs). TFs, miRNAs and their target gene(s) are indicated in triangles, rounded rectangles and circles, respectively. Genes upregulated and down-regulated in all patient tumor samples are indicated in red and blue, respectively. Purple edges indicate TF–target regulation, while red edges indicate miRNA–target regulation. (**D**) The miR-9-5p–TF gene regulatory network. (**E**) The miR-9-5p–FOXO1–CPEB3 FFL.

**Figure 2 cells-11-02116-f002:**
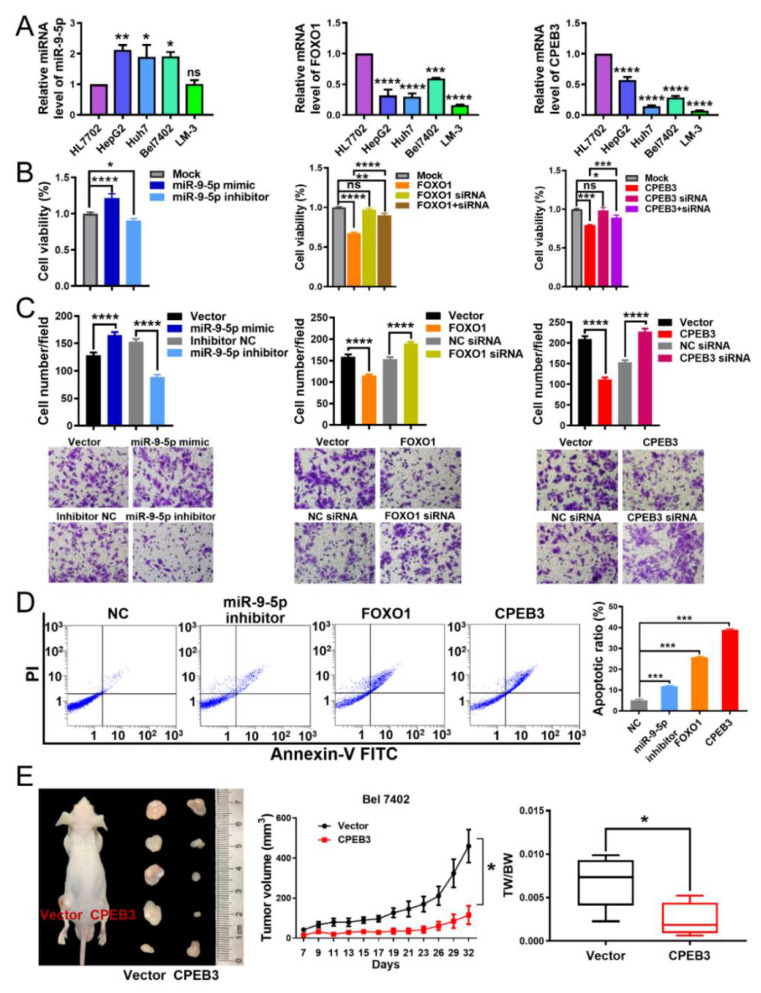
miR-9-5p promotes the growth of HCC cells in vitro; FOXO1 and CPEB3 inhibit cell growth. (**A**) qRT-PCR expression analysis of miR-9-5p, *FOXO1* and *CPEB3* expression in liver (HL7702) and HCC cell lines. Gene expression levels in cell lines were calculated as relative expression to HL7702 (normalized to 1). Non-parametric one-way ANOVA with Dunnett’s multiple comparison post hoc test was performed. Mean ± s.e.m., *n* = 3/group. Unless otherwise stated, the subsequent tests were carried out in the same way. (**B**) Cell viability of Bel7402 cells after transfection with Mock (control cells group), miR-9-5p mimics, inhibitor as indicated (left); Transfected with Mock, *FOXO1* siRNA or *FOXO1* vector with or without siRNA (middle); Transfected with Mock, *CPEB3* siRNA or *CPEB3* vector with or without siRNA (right). Cell viability in treatment groups was calculated as relative values to Mock (normalized to 1). *n* = 4/group in cell viability analyses, while *n* = 9/group in cell number analyses. (**C**) Transwell assays of cellular invasion after transfection with miR-9-5p mimic or its inhibitor, *FOXO1* or *FOXO1*-specific small interference RNA (siFOXO1), and *CPEB3* or *CPEB3*-specific small interference RNA (siCPEB3) in Bel7402 cells. Representative images (bottom) and quantification of 10 randomly selected fields (top) are shown. (**D**) Apoptosis in Bel7402 cells induced by miR-9-5p inhibitor, FOXO1, and CPEB3 was detected using flow cytometry. *n* = 3/group. (**E**) The in vivo effect of CPEB3 was evaluated in xenograft mouse models bearing tumors originating from Bel7402 cells. Tumor volume and tumor weight (TW) was periodically measured for each mouse, and tumor growth curves and box plot were plotted. One-way ANOVA was used for significance test of tumor volume and Student’s *t*-test for TW. Mean ± s.e.m., *n* = 6/group. * *p* < 0.05, ** *p* < 0.01, *** *p* < 0.001, **** *p* < 0.0001, ns: non-significant.

**Figure 3 cells-11-02116-f003:**
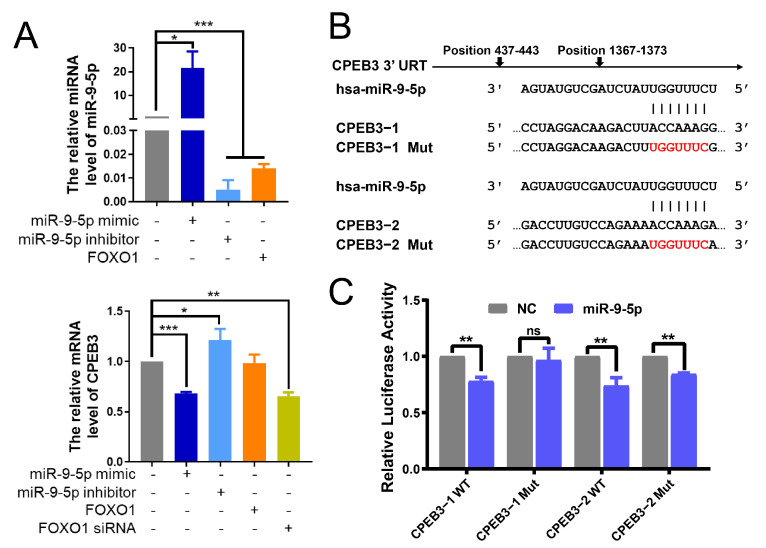
miR-9-5p is a negative regulator of *CPEB3*. (**A**) Levels of miR-9-5p, *CPEB3* were detected by qRT-PCR in miR-9-5p mimic, miR-9-5p inhibitor or FOXO1, siFOXO1 treated Bel7402 cells. Relative expression levels were calculated for treatment groups compared to the control group (normalized to 1), by non-parametric one-way ANOVA and Dunnett’s multiple comparison post hoc tests. Mean ± s.e.m, *n* = 3/group. (**B**) Schematic representation of the 3′ UTR regions of *CPEB3* (*CPEB3* transcript variant 2, NCBI: NM_001178137.1) with the putative miR-9-5p binding sites (CPEB3−1, position 437-443 of *CPEB3* 3′ UTR; CPEB3−2, position 1367-1373 of *CPEB3* 3′ UTR), including wild-type (CPEB3−1 WT, CPEB3−2 WT) and mutant (CPEB3−1 Mut−1, CPEB3−2 Mut−2). (**C**) Relative luciferase activity of the indicated *CPEB3* 3′ UTR vectors in 293T cells transfected with miR-9-5p plasmids. While luciferase activity of control group was normalized to 1, Renilla luciferase activity was normalized to firefly activity and is presented as relative luciferase activity. Student’s *t*-test. Mean ± s.e.m., *n* = 3/group. * *p* < 0.05, ** *p* < 0.01, *** *p* < 0.001, ns: non-significant.

**Figure 4 cells-11-02116-f004:**
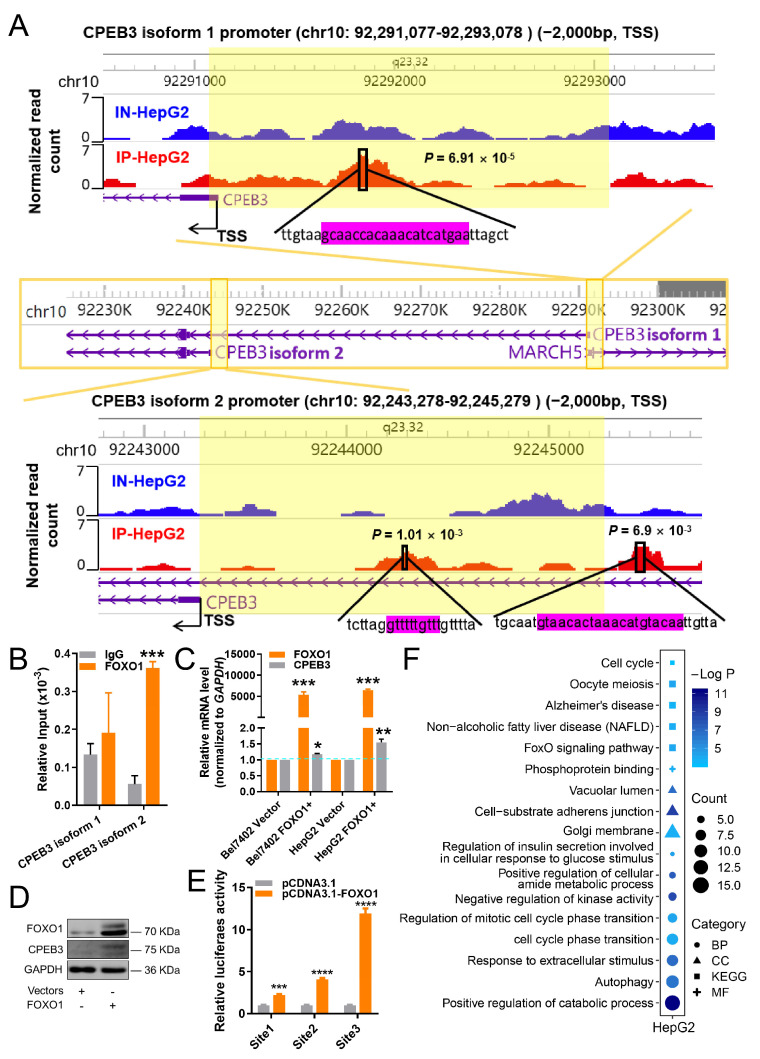
FOXO1 is a positive regulator of *CPEB3*. (**A**) ChIP-Seq tag profiles for FOXO1 normalized read count at the *CPEB3* promoters of two isoforms (NCBI: NM_014912.5 for isoform 1, NM_001178137.1 for isoform 2) in HepG2 cells under no-immunoprecipitation (IN) or immunoprecipitation (IP) conditions. Locations of FOXO1-binding sites (magenta color) are indicated. *p*-value of each binding site was calculated by MACS2. Light yellow indicates the promoter region of *CPEB3*. TSS, transcription start site. (**B**) Estimation of ChIP-Seq via quantitative ChIP-PCR in HepG2 cells normalized by input control, IgG served as a negative control. *CPEB3* isoforms 1 and 2 were the same as shown in (**A**). Mean ± s.e.m., *n* = 3/group. (**C**) Gene expression levels of *FOXO1* and *CPEB3* were quantified by qRT-PCR as relative expression to control (vector). Expression of control was normalized to 1. Student’s *t*-test. Mean ± s.e.m., *n* = 3/group. (**D**) Protein expression of FOXO1 and CPEB3 was assessed by Western blot assay in HepG2 cells. (**E**) HepG2 cells were transfected with three sites of *CPEB3* promoter with firefly and Renilla luciferase reporter constructs and plasmids, respectively, as indicated. At 48 h after transfection, renilla luciferase activity was normalized to firefly activity and is presented as relative luciferase activity, as described in “Supplementary Methods”. Site 1 indicates the binding site of FOXO1 to *CPEB3* isoform 1 in (**A**), while site 2 and site 3 indicate the binding sites of isoform 2 of *CPEB3* ((**A**), bottom, left and right sites, respectively). Student’s *t*-test. Mean ± s.e.m., *n* = 3/group. (**F**) Significantly enriched terms (*p* < 0.01) in Gene Ontology and KEGG pathway for DEGs of *FOXO1* overexpression compared with control in HepG2 cells. CC: cellular component. BP: biological process. MF: molecular function. * *p* < 0.05, ** *p* < 0.01, *** *p* < 0.001, **** *p* < 0.0001, ns: non-significant.

**Figure 5 cells-11-02116-f005:**
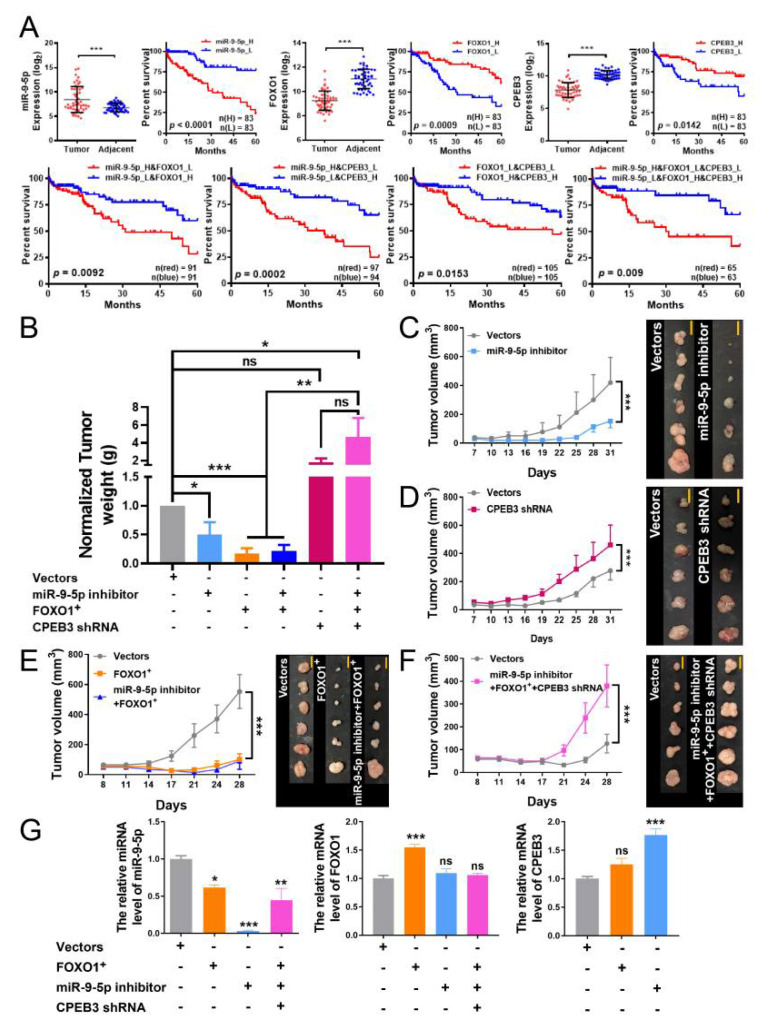
Effects of miR-9-5p/FOXO1/CPEB3 FFL on survival and tumor growth in vivo. (**A**) Boxplots indicate the expression of miR-9-5p, *FOXO1* and *CPEB3* in TCGA paired HCC and adjacent tissue samples (paired *t*-test). Kaplan–Meier survival analysis of single genes or combinations of two or three genes in miR-9-5p/FOXO1/CPEB3 FFL at 60 months (log-rank test) in HCC patients. L: Low-expression samples, H: High-expression samples. (**B**) Tumor formation experiment with different treated Bel7402 cells. Tumor weights were detected in the various combinations of miR-9-5p inhibitor/*FOXO1*/*CPEB3*-shRNA through tumor hepatoma xenografts. Different combinations have corresponding vectors (blank controls) and all are normalized to 1g as shown in the grey color bar. Normalized tumor weight of treated group is calculated as relative to control (before normalized). Tumor volume of miR-9-5p inhibitor (**C**), *CPEB3* shRNA (**D**), *FOXO1* or *FOXO1* and miR-9-5p inhibitor combination (**E**) and three genes miR-9-5p inhibitor, *FOXO1* and *CPEB3* shRNA combination (**F**) were periodically measured for each mouse, and growth curves were plotted. Scale bar = 1 cm. Significance test in B was performed with Student’s *t*-test, others were one-way ANOVA. Mean ± s.e.m., *n* = 6/group. (**G**) Levels of miR-9-5p (left), *FOXO1* (middle), *CPEB3* (right) were detected by qRT-PCR in tumor tissues in (**B**). Relative expression levels were calculated for treatment groups compared to the controls (normalized to 1). Non-parametric one-way ANOVA and Dunnett’s multiple comparison post hoc tests. Mean ± s.e.m, *n* = 3/group. * *p* < 0.05, ** *p* < 0.01, *** *p* < 0.001, ns: non-significant.

**Figure 6 cells-11-02116-f006:**
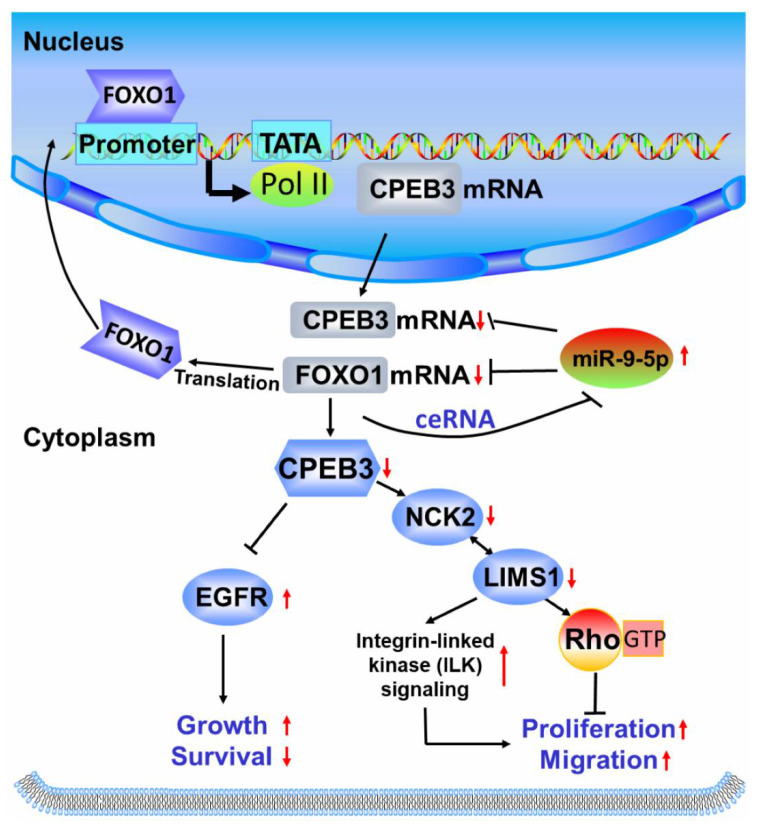
A potential pathway mediated by miR-9-5p/FOXO1/CPEB3 FFL. Schematic illustration of miR-9-5p/FOXO1/CPEB3 FFL and its possible downstream pathways in hepatocellular carcinoma. Up arrows represent increased gene expression (*CPEB3*, *FOXO1* etc.) and indicated pathway effects (tumor growth, proliferation etc.), while down arrows represent decreased gene expression and indicated pathway effects. ceRNA: competing endogenous RNA.

## Data Availability

The sequencing data of RNA-Seq and ChIP-Seq of FOXO1 overexpressed in HepG2 cells presented in this study are openly available in National Genomics Data Center, reference number CRA003225, which is publicly accessible at https://ngdc.cncb.ac.cn.

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
