# Peer review of "A miR-9-5p/FOXO1/CPEB3 Feed-Forward Loop Drives the Progression of Hepatocellular Carcinoma"

_cells, 2022, doi:10.3390/cells11132116_

Round 1
Reviewer 1 Report
I think the authors did a quite comprehensive analysis.
The authors newly identified the critical feed-forward loop (FFL) regulatory module (miR-9-5p/FOXO1/CPEB3 FFL) was associated with poor prognosis and promoted cell growth and tumor progression of HCC in vitro and in vivo.
Comments
1. Supplementary data (Figures and Table S1, Table S3) are missing.
2. Clinical characteristics of these 166 patients from TCGA in Figure 1, such as the distribution of their etiology, tumor stages, and any treatment?
3. To further confirm their results, I would suggest a Ki67 staining in vivo, to verify the statement of "tumor proliferation" is influenced by the loop. 4. Besides, distribution of pathological grades in vivo would be of help in clinical application. 5. Figure 1F can be combined into Figure 5A. 6. Can authors explain the findings in Figure 2A that why LM-3 HCC cells had low mRNA of mir-9-5p but still had very low level of CPEB3 and FOXO1? Does other possible molecules, other than miR-9-5-p regulated loop, involved in growth?
Author Response
We would like to express our sincere thanks to the reviewers for the constructive and positive comments. We have revised the manuscript according to all the reviewers’ suggestions.
We have a point-by-point response and the changes in the revised manuscript have been marked in red.
Please see the attachment.

Reviewer 2 Report
The Authors have conducted an interesting mechanicistic study on HCC. I feel that the paper is intended to an audience of molecular biologists. However, can the Authors expand on the relevance of their findings to clinical practice?
- Abstract, please expand on the rationale for the study
- Introduction, please do not "spoiler" the findings of the study in the last paragraph, but provide only the aim of the study. futhermore, the first paragraphs of the introduction can be omitted.
- Discussion, can the Authors expand on the further studies that can be conducted upon the results of this one?
- Discussion, can the Authors comment on the potential limitations of their study?
Author Response

(The authors gave the same response as above.)

Reviewer 3 Report
This study shows a miR-9-5p/FOXO1/CPEB3 feedforward loop that acts to the progression of HCC. The authors have combined bioinformatic data with quite a few delicate experiments and exhibited convincing data. Nevertheless, some points need to be addressed.
Major points:
1. Introduction: The statement of the concept "feedforward loop" requires more elaboration.
2. What assay exactly is shown in Figure 2C? Migration or Invasion? In the Materials and Methods section, both migration and invasion assays are described.
4. Materials and Methods: Please specify how you conducted the apoptosis assay and the way you defined apoptosis % in a flow cytometry.
5. Materials and Methods: Specify the procedures of xenograft tumor model. Grafted cell type? Number? Time? Any operational criteria?
6. Materials and Methods: Please state the approval number and date of the IACUC.
Minor points
Specify the full names when the abbreviations show up the first time. E.g., line 46, 49, 52, Figure 6 and the legend... and so on.
Figure 6: ceRNA?
Author Response
We would like to express our sincere thanks to the reviewers for the constructive and positive comments. We have revised the manuscript according to all the reviewers' suggestions.
We have a point-by-point response and the changes in the revised manuscript have been marked in red.
Please see the attachment.

Round 2
Reviewer 3 Report
The explicit revisions to all the issues are appreciated. From my perspective, this article is qualified for acceptance.